



# Potential predictability of marine ecosystem drivers

Thomas L. Frölicher[1,2], Luca Ramseyer[1], Christoph C. Raible[1,2], Keith B. Rodgers[3,4], John Dunne[5]

[1]Climate and Environmental Physics, Physics Institute, University of Bern, Bern, 3012, Switzerland
[2]Oeschger Centre for Climate Change Research, University of Bern, Bern, 3012, Switzerland
[3]Centre for Climate Physics, Institute for Basic Science, Busan, South Korea
[4]Pusan National University, Busan, South Korea
[5]NOAA Geophysical Fluid Dynamics Laboratory, Princeton, NJ, USA

*Correspondence to*: Thomas L. Frölicher (thomas.froelicher@climate.unibe.ch)

**Abstract.** Climate variations can have profound impacts on marine ecosystems and the socio-economic systems that may depend upon them. Temperature, pH, oxygen ($O_2$) and net primary production (NPP) are commonly considered to be important marine ecosystem drivers, but the potential predictability of these drivers is largely unknown. Here, we use a comprehensive Earth system model within a perfect modelling framework to show that all four ecosystem drivers are potentially predictable
on global scales and at the surface up to 3 years in advance. However, there are distinct regional differences in the potential predictability of these drivers. Maximum potential predictability (>10 years) is found at the surface for temperature and $O_2$ in the Southern Ocean and for temperature, $O_2$ and pH in the North Atlantic. This is tied to ocean overturning structures with 'memory' or inertia with enhanced predictability in winter. Additionally, these four drivers are highly potentially predictable in the Arctic Ocean at surface. In contrast, minimum predictability is simulated for NPP (<1 years) in the Southern Ocean.
Potential predictability for temperature, $O_2$ and pH increases with depth to more than 10 years below the thermocline, except in the tropical Pacific and Indian Ocean, where predictability is also three to five years in the thermocline. This study indicating multi-year (at surface) and decadal (subsurface) potential predictability for multiple ecosystem drivers is intended as a foundation to foster broader community efforts in developing new predictions of marine ecosystem drivers.

## 1 Introduction

Marine organisms and ecosystems are strongly influenced by seasonal to decadal-scale climate variations, challenging the sustainable management of living marine resources (Drinkwater et al., 2010; Lehodey et al., 2006). Anomalies in temperature, pH, $O_2$ and nutrients are important drivers of such climate-induced ecosystem variations (Gattuso et al., 2015; Gruber, 2011). Therefore, skillful predictions of these marine ecosystem drivers have considerable potential for use in marine resource management (Gehlen et al., 2015; Hobday et al., 2016; Payne et al., 2017; Tommasi et al., 2017).

The primary tools for investigating how marine organisms and ecosystems change on seasonal to decadal timescales are Earth system models, where prognostic equations are implemented for biogeochemical cycles. These models are capable of



representing both natural variability and transient changes in the marine ecosystem drivers (Bopp et al., 2013; Frölicher et al., 2016). Recently, Earth system models have been used to explore and quantify the predictability of marine biogeochemical
tracers. Most of the studies focus on predicting the ocean uptake of carbon (Li et al., 2016, 2019; Lovenduski et al., 2019; Séférian et al., 2018).

To date, only a few studies have investigated the predictability of marine ecosystem drivers (Chikamoto et al., 2015; Park et al., 2019; Séférian et al., 2014a). An intriguing finding of these studies is that marine biogeochemical drivers may be more
predictable than their physical counterparts. Séférian et al. (2014a), for example, showed that net primary productivity (NPP) has greater predictability than sea surface temperature (SST) in the eastern equatorial Pacific. They hypothesized that SST is strongly influenced by high-frequency surface fluxes, whereas NPP is more directly impacted by thermocline adjustment processes that determine the rate at which nutrients are brought into the ocean's euphotic layer. Thus, biogeochemical predictions may hold great promise and highlight the need for further investigation. Changes in ecosystem drivers have impacts
not only on the surface ocean, but over upper ocean waters spanning the euphotic zone and below making it important to understand more broadly how ecosystem drivers vary over a range of depths. To our knowledge there is no comprehensive assessment of potential predictability of marine ecosystem drivers at the global scale spanning multiple depth horizons, and a comparison of the relative predictability among them.

In this study, we assess the potential predictability of the four marine ecosystem drivers using 'perfect model' simulations of a comprehensive Earth system model. We address following three questions:

- To what extent are marine ecosystem drivers predictable at the global scale?
- What are the regional and depth-dependent characteristics of potential predictability?
- Which underlying physical and biogeochemical processes prescribe or limit the potential predictability of marine
ecosystem drivers?

This study is organized as follows. First, we introduce the model and methods used to assess the potential predictability in marine ecosystem drivers. Subsequently, the temporal sequencing of potential predictability over global scales for the four marine ecosystem drivers are identified and evaluated for regional differences in potential predictability horizons. Both surface
and subsurface manifestations are presented to assess the origin of potential predictability. Finally, we also identify the mechanistic controls on the limits to potential predictability and conclude with a discussion and summary section.



## 2 Methods

### 2.1 Earth system model: GFDL ESM2M

For this study we conducted a new 240-member ensemble suite of simulations of 10-year duration each with the Earth system model ESM2M developed at the Geophysical Fluid Dynamics Laboratory (GFDL) of the National Oceanic and Atmospheric Administration (NOAA) (Dunne et al., 2012, 2013). The GFDL ESM2M is a fully coupled carbon cycle-climate model. The physical core of the model is based on the physical coupled model CM2.1 (Delworth et al., 2006). The atmospheric model AM2 has a horizontal resolution of 2° latitude × 2.5° longitude with 24 vertical levels (Anderson et al., 2004). The land model simulates land water, energy and carbon cycle, and has the same horizontal resolution as the atmospheric component. The ocean model MOM4p1 (Griffies, 2012) has 50 vertical levels of varying thickness and a nominal horizontal resolution of 1° latitude × 1° longitude, increasing towards the equator to up to 1/3°. The sea ice model includes full ice dynamics, three thermodynamic layers and five ice thickness categories and is defined on the same grid as the ocean model (Winton, 2000).

Ocean biogeochemistry and ecology is simulated by the Tracers Of Phytoplankton with Allometric Zooplankton version 2.0 (TOPAZ2) (Dunne et al., 2013). TOPAZ2 represents 30 prognostic tracers to describe the cycles of carbon, phosphorus, silicon, nitrogen, iron, alkalinity, oxygen and lithogenic material as well as surface sediment calcite. TOPAZ2 includes three phytoplankton functional groups: small (mostly prokaryotic pico- or nanoplankton), diazotroph (fixing nitrogen from the atmosphere), and large phytoplankton. TOPAZ2 only implicitly simulates zooplankton activity. The growth of phytoplankton depends on the level of photosynthetically active irradiance, nutrients (e.g. nitrate ammonium, phosphate, and iron) and temperature (see section 2.3.2 and Appendix A).

Previous studies have shown that the GFDL ESM2M captures the observed large-scale biogeochemical patterns (Dunne et al., 2012, 2013). The GFDL CM2.1 skillfully simulates primary modes of natural climate variability (Wittenberg et al., 2006), and has been extensively applied to assess seasonal and multiannual climate predictions (Meehl et al., 2013; Park et al., 2019).

### 2.2 Perfect model framework

We estimated potential predictability within a 'perfect model' experiment. A 300-yr preindustrial control simulation (black line in Figure 1) is performed, which is branched off a pre-existing quasi-steady-state 1000-yr preindustrial control simulation. Using this 300-yr preindustrial control simulation to provide initial conditions, six 40-member ensemble simulations of 10-yr duration each are performed. Each ensemble simulation starts at different times in the control simulation: January 1$^{st}$ in years 22, 64, 106, 170, 232 and 295, respectively. Note that the last ensemble exceeds the control simulation by 5 years. The six starting years of the individual large ensemble simulations are randomly selected from the 300-yr preindustrial control simulation. Each of the six ensembles consists of 40 ensemble members with micro-perturbations to oceanic initial states but with the same atmospheric, land, ocean biogeochemical, sea ice, and iceberg initial conditions. Specifically, for each ensemble





member, $i = 1, 2, \ldots, 40$, an infinitesimal temperature perturbation $\delta$ is added to a single grid cell in the Weddell Sea at 5-m depth, similar to the approach described in Wittenberg et al. (2014a) and Palter et al. (2018):

$$\delta_i = 0.0001°C \times \begin{cases} \frac{i+1}{2}: & for\ odd\ i \\ -\frac{i}{2}: & for\ even\ i \end{cases}.$$  (1)

As stated above, our model setup encompasses 240 ensemble members, each of 10-yr duration and thus 2400 yr of model integration in addition to the 300-yr long control simulation. While our perturbation method is in no way optimal in terms of, for example, sampling the likely range of atmospheric-ocean-biogeochemical errors, it is sufficient to generate ensemble spread on the timescales of interest. After just four days of simulation time subsequent to the micro-perturbations for each cluster of 40 starting points, the SST of all surface ocean grid cells are numerically different from the SST of the control

simulation, underscoring the rapidity with which divergences due to nonlinearities in the model express themselves. The method applied here mirrors that of Griffies and Bryan (1997a), Msadek et al. (2010), and Wittenberg et al. (2014b), and emphasizes the amplitude (but not the phase) of perturbations to identify potential predictability. Our perturbation method produces ensemble experiments likely to give the upper limit of the model predictability, hence the term potential predictability. Nevertheless, it warrants mentioning here that studies have been published arguing that predictability in the real

world for some variables may even be larger than estimated with the perfect modeling framework within an Earth system model in cases where the ratio of the predictable mode to model noise is underrepresented (Eade et al., 2014; Kumar et al., 2014).

**2.3 Analysis methods**

We calculate the potential predictability for the four marine ecosystem drivers: temperature, pH, $O_2$ and NPP. In the following,

NPP is always integrated over the upper 100 m, whereas temperature, pH and $O_2$ are analyzed at different depth levels. In addition to identifying the upper limits of predictability of these variables within the Earth system model, an equally important objective is to identify the relative predictability of the four variables under consideration.

**2.3.1 Assessment of potential predictability**

The prognostic potential predictability (PPP) is the main metric used in this study to assess predictability. The PPP is the ratio

between the variance among the ensemble members at a given time $t$ after the initialization and the temporal variance of an undisturbed control simulation. The PPP is calculated following Griffies and Bryan (1997b) and Pohlmann et al. (2004):

$$PPP(t) = 1 - \frac{\frac{1}{N(M-1)}\sum_{j=1}^{N}\sum_{i=1}^{M}(X_{ij}(t)-\bar{X}_j(t))^2}{\sigma_c^2}$$  (2)





where $X_{ij}$ is the value of a given variable for the $j$-th ensemble and $i$-th ensemble member, $\bar{X}_j$ is the mean of the $j$-th ensemble

over all ensemble members, $\sigma_c^2$ is the variance of the control simulation, $N$ is the total number of different ensemble simulations

($N = 6$) and $M$ the number of ensemble members ($M = 40$). The variance of the control simulation is calculated for each month

of the year separately to exclude the seasonality from the natural variability, i.e., only the natural variability at that month in

the seasonal cycle is considered. PPP equals unity constitutes perfect predictability. A F-test is applied to estimate a significant

difference between the ensemble variance and the variance of the control run. With $N = 6$ and $M = 40$, predictability is achieved

with a 95% confidence level when PPP $\geq 0.183$.

The predictability time horizon is defined as the lead time at which PPP falls below the predictability threshold. To calculate

global means, all metrics are first calculated at each individual grid cell and then averaged with area-weighting over the global

ocean.

**2.3.2 Taylor deconvolution method to identify mechanistic controls of predictability**

The prognostic

To understand the processes behind the simulated predictability, we applied a first-order Taylor-series deconvolution method

to decompose the normalized ensemble variance of pH, $O_2$ and NPP into contributions from their physical and biogeochemical

driver variables:

$$\sigma_f^2 \cong \sum_{i=1}^{n}\left(\frac{\partial f}{\partial x_i}\Big|\sigma_{x_i}\right)^2 + 2\sum_{i<j}\frac{\partial f}{\partial x_i}\Big|\frac{\partial f}{\partial x_j}\Big| Cov(x_i, x_j) , \qquad (3)$$

where $\sigma$ denotes the standard deviation among the ensemble members of the different variables. Specifically, the Taylor

deconvolution method is applied to decompose the normalized ensemble variance for $f$ being of pH, $O_2$ and NPP into the

contribution from their physical and biogeochemical drivers by expressing the ensemble variance and the variance of the

control run from equation (2) in terms of equation (3). The partial derivatives in equation (3) are calculated at the point $\vec{p} = \vec{x}$,

where $\vec{x}$ is the mean value of the corresponding driver variables over the entire control simulation.

The changes in pH (or $H^+$) are attributed to changes in temperature, salinity, total alkalinity (Alk), and total dissolved inorganic

carbon (DIC). Here, we assume that variations in phosphate and silicate are negligible.

Dissolved oxygen ($O_2$) is decomposed into an oxygen solubility component $O_2^{sol}$ and an apparent oxygen utilization (AOU)

component using (e.g. Frölicher et al. 2009):




$$O_2 = O_2^{\text{sol}} - AOU. \tag{4}$$

$O_2^{\text{sol}}$ is the solubility of oxygen, which depends non-linearly on temperature and salinity (Garcia and Gordon, 1992). The difference between diagnosed $O_2^{\text{sol}}$ and simulated $O_2$ is AOU. Variations in AOU reflects changes in oxygen consumption and ocean ventilation. Earlier studies demonstrated that changes in AOU are typically associated with changes in ventilation, as simulated changes in the remineralization rates of organic material and in associated $O_2$ consumption are relatively small (Gnanadesikan et al., 2012).

NPP can be decomposed into the contributions from the three phytoplankton groups simulated in the TOPAZ model:

$$\text{NPP} = \text{NPP}_{\text{Sm}} + \text{NPP}_{\text{Di}} + \text{NPP}_{\text{Lg}} \tag{5}$$

where $\text{NPP}_{\text{Sm}}$, $\text{NPP}_{\text{Di}}$, and $\text{NPP}_{\text{Lg}}$ are the contributions from small, diazotroph and large phytoplankton, respectively. At any time $t$ the NPP for all phytoplankton groups *phyto* is given by the phytoplankton stock $P_{phyto}$ times the phytoplankton growth rate $\mu_{phyto}$:

$$\text{NPP}_{phyto}(t) = \mu_{phyto}(t) \cdot P_{phyto}(t) \tag{6}$$

The growth rate $\mu_{Sm}$ of the small phytoplankton is parametrized using a maximum growth rate $\mu_{max}$, which is limited by nutrients $N_{lim}$, light $L_{lim}$, and temperature $T_f$ (see Appendix A for further details):

$$\mu = \mu_{max} \cdot N_{lim} \cdot L_{lim} \cdot T_f \,. \tag{7}$$

Note that grazing, sinking and other loss processes impact phytoplankton stock, but these processes in TOPAZ2 are only a function of steady state growth and biomass implicit grazing formulation, and exert no separate dynamic control. Therefore they do not require separate consideration.

## 3 Results

### 3.1 Potential predictability at the ocean surface

The change in globally averaged annual PPP with lead time is very similar for all four marine ecosystem drivers at the surface, i.e. the PPP decreases exponentially over lead time for all four drivers (solid thick lines in Figure 2). After three years, the PPP falls below the predictability threshold (dashed line in Figure 2) indicating that the global predictability is about three years



for all four ecosystem drivers. The seasonality in PPP (solid thin lines in Figure 2) as well as the differences among the four drivers is very small at the global scale.

At the regional scale, the predictability time horizon shows distinct structured patterns and also large differences across the four different marine ecosystem drivers (Figure 3). In general, SST (Figure 3a), surface pH (Figure 3b) and surface $O_2$ (Figure 3c) share similar predictability patterns with low predictability (1-2 years) between 20° and 40° in both hemispheres, moderate predictability (3-5 years) in the tropical oceans, and high predictability (>10 years) in the North Atlantic between 40°N and 70°N, in the Southern Ocean between 40°S and 65°S (except for surface pH), and in the Arctic Ocean. Interestingly, potential

predictability of surface pH is low relative to SST and surface $O_2$ in the Southern Ocean, but elevated over both the Caribbean and the eastern subtropical North Pacific relative to SST. The Caribbean and the eastern North Pacific are both regions of importance for resource management, given the high density of neighboring human populations.

The NPP predictability pattern (Figure 3d) is fundamentally different from the patterns of the other three ecosystem drivers.

NPP has high predictability (6-10 years) in the mid-latitudes, where the annual mean NPP is generally small (indicated with contour lines in Figure 3d), but very low predictability of 0-1 years in the Southern Ocean, the North Atlantic and the Pacific, as well as low predictability of 1-3 years in the tropical oceans, where annual mean NPP is high (Figure 3d). The spatial pattern of predictability and the sequencing of predictability among the ecosystem drivers is very similar when using two other metrics for potential predictability indicating that our results do not depend on the predictability metric used (Appendix B).


We further average the local potential predictability across 17 biogeographical biomes (Figure 4) to highlight the pronounced seasonal cycle in predictability for some variables in particular biomes. The biomes capture patterns of large-scale biogeochemical function at the basin scale and are defined by distinct SSTs, maximum mixed layer depths, maximum ice fractions, and summer chlorophyll concentrations (Fay and McKinley, 2014). As shown in Figure 4, potential predictability

exhibits strong seasonality for SST, surface $O_2$ and surface pH in the North Atlantic (biomes 8, 9, 10, 11), in the Southern Ocean (biomes 15 and 16), and in the subtropical/subpolar gyre boundary region of the North Pacific (biome 3). In all these biomes, predictability is higher during the cold season (boreal and austral winter) and lower during the warm season. The biomes with high seasonality in PPP are also the regions which generally show larger predictability in the annual mean. The PPP of SST and surface $O_2$ have almost identical seasonal amplitudes, while the seasonal amplitude of the surface pH is

generally smaller compared to SST and surface $O_2$ seasonal amplitude. Interestingly, the PPP for NPP generally shows no large differences amongst the seasons, except in biome 8, which is influenced by seasonal sea-ice retreat/growth. Figure 4 reveals also other interesting characteristics in PPP. For example, the changes in PPP over lead time are very small, but fluctuate around the predictability threshold for NPP in biome 10 and for SST and $O_2$ in biome 8, making the predictability horizon in some biomes for some variables very sensitive to small changes in PPP. In addition, the PPP for NPP in the eastern

equatorial Pacific (biome 6) shows large interannual variations with lead time indicating that even more ensemble members





are needed to robustly assess the predictability there. The PPP for SST in biome 17 (around Antarctica) is even negative indicating a higher variance simulated in the ensemble simulations than simulated in the 300-yr preindustrial control simulation.

**3.2 The role of the subsurface ocean in the potential predictability of marine ecosystem drivers**

Next, we assess the potential predictability for temperature, $O_2$ and pH in the top 1000 m (Figures 5 and 6). In theory, the subsurface ocean should be expected to be more predictable than the surface layer, as the subsurface is not directly coupled to the high-frequency and relatively unpredictable variability of the atmosphere. Indeed, the potential predictability for temperature, oxygen and pH rapidly increases with depth at the global scale (Figure 5a-c). Below 300 m, the predictability of all three ecosystem drivers exceeds a decade, i.e. the PPP is still larger than the predictability limit (depth levels with no

hatching in Figure 5a-c). Interestingly, the PPP at depth changes more rapidly with time for temperature than for $O_2$ and pH. In fact, the PPP for temperature is constant below 500 m for a given year, i.e. the PPP value does not change with depth. This is different for $O_2$ and pH, for which the PPP increases with all depth levels. Clearly, the overall increasing potential predictability with depth can be attributed to the increasing disconnection of the deeper ocean with the surface ocean (see also section 3.3). However, the coupling between physical and biogeochemical processes leads to an enhanced predictability below

500 m, this being the case for oxygen and pH, but not the case for temperature.

The global mean picture of Figure 5a-c obscures some interesting seasonal features at the regional scale, which are highlighted in Figure 5d-f for the North Atlantic. Even though the North Atlantic is among the regions with the largest potential predictability at the ocean surface, the predictability at 1000-m depth for pH and $O_2$ is smaller in the North Atlantic than the

global average at the same depth (Figure 5d-f), especially in boreal winter. For example, the PPP in winter of year 3 for pH is 0.6 at the global scale at 400-m depth (Figure 5b), but only 0.3 in the North Atlantic (Figure 5e). The strong connection in the Atlantic between the ocean surface and the upper 1000 m in winter increases the predictability, but at the same time, decreases the potential predictability within the subsurface. Interestingly, this effect is also visible for temperature but confined to the upper few hundred meters. The reason is that anomalies from the ocean surface do not penetrate as deep for pH and $O_2$ as they

do for temperature.

Figure 6 shows the spatial pattern of the predictability time horizon for ocean temperature, $O_2$ and pH at 300 m (a-c) and 1000 m (d-f) depth, respectively. Although the potential predictability is close to 10-yr lead time below 300 m on global average, there are specific regions with a reduced prediction horizon. At 300 m, these regions are the tropical Pacific, the Indian Ocean and parts of the Southern Ocean (Figure 6a-c). In the equatorial Pacific and Indian Ocean averaged over 20°N and 20°S, the

predictability is 4 yr for temperature and 7 yr for $O_2$ and pH, respectively. For temperature and $O_2$, potential predictability drops to lead times lower than 5-6 yr in the eastern equatorial Atlantic. At 1000-m depth (Figure 6d), the spatial pattern of temperature predictability is similar to the one at 300 m. Large parts of the equatorial Pacific and the Indian Ocean still show





relatively low predictability. This is not the case for O₂ and pH, for which the predictability largely increases at 1000-m depth
compared to 300 m depth in the eastern equatorial Pacific and in the Indian Ocean as well as in the Southern Ocean, so that
both O₂ and pH are almost everywhere predictable up to 10-yr lead time. Only the western equatorial Pacific (for pH) and the
central equatorial Pacific (for O₂) are characterized by reduced potential predictability at 1000 m (lead times lower than 8 yr).

### 3.3 Deconvolution into physical and biogeochemical control processes

The predictability patterns and timescales presented in the previous sections are investigated next for their underlying
dynamical and/or biogeochemical controls. For SST, we compare our findings with previous studies that attributed SST
predictability to particular processes. In order to understand the dynamical and biogeochemical control processes of O₂, pH
and NPP and to quantify their contribution, we apply a Taylor deconvolution method (see section 2.3.2). It is important to note
that large contribution of a particular driver to the potential predictability of O₂, pH and NPP does not imply a high
predictability of that driver. In addition, the contribution of a process depends not only on its potential predictability (captured
by the variance terms in equation 3), but also on the potential interaction with the other drivers (covariance terms in equation
3).

### 3.3.1 Sea surface temperature

The high predictability of SST in the North Atlantic between 40°N and 70°N (Figure 3a) is consistent with previous findings
(Boer, 2004; Collins et al., 2006; Griffies and Bryan, 1997a; Pohlmann et al., 2004). The SST in the North Atlantic experiences
low-frequency variability that is linked to the Atlantic Meridional Overturning Circulation (AMOC, Buckley and Marshall
(2016)). In fact, the AMOC has a significant peak in its power spectrum at 20 yr in GFDL's CM2.1 (Msadek et al., 2010), the
physical core of the GFDL ESM2M used here. Similarly, the Southern Ocean surface waters are also strongly connected to
the deep ocean (Morrison et al., 2015) and slow subsurface ocean processes there give rise to decadal predictability in SST
(Marchi et al., 2019; Zhang et al., 2017). In CM2.1, the peak in the power spectrum of deep convection in the Weddell Sea is
simulated to lie between 70 and 120 years (Zhang et al., 2017). In the North Atlantic and the Southern Ocean, the potential
predictability is enhanced during the winter period (Figure 4), as the surface waters are especially well connected with the
deep ocean during the cold season. The high SST predictability in the Arctic Ocean is due to the overall low-frequency
variability in SST there, because these waters are permanently covered by sea ice in the preindustrial ESM2M control
simulation and cannot exchange heat (and carbon) with the atmosphere. This is not the case around the Antarctic continent,
where sea ice almost vanishes during austral summer in ESM2M allowing the surface ocean to exchange heat and carbon with
the atmosphere. Therefore, the influence of high frequency atmospheric variability is large leading to small predictability
around Antarctica. Moderate predictability in SST of about 3 to 5 years is simulated in the tropical oceans associated with the
coupled atmosphere-ocean system (Boer, 2004) .



### 3.3.2 Dissolved oxygen

To understand the processes that give rise to the $O_2$ predictability pattern, we use a Taylor deconvolution method (see section 2.3.2) to further split the $O_2$ predictability into respective $O_2^{sol}$ and an AOU contributions. Figures 7 and 8 show the predictability time horizon of $O_2$ (identical to patterns shown in Figures 3c and 6c), $O_2^{sol}$, AOU and their covariance (left panels) as well as their percentage contribution to the normalized ensemble variance (right panels) for the surface (Figure 7) and 300-m depth (Figure 8). The percentage contribution is defined as the value of a given variance term (first term on the

right hand side of the equal sign in equation 3) or covariance term (second term on the right hand side in equation 3), divided by the sum of all absolute variance and covariance values. By combining the information from the right panels (i.e. percentage contribution to total predictability) with the information from the left panels (i.e. predictability time horizon), we can attribute the local predictability of $O_2$ to either $O_2^{sol}$, AOU or the covariance. For example, if both the percentage contribution as well as the predictability time horizon of particular variable is high, then the $O_2$ predictability is high. If the percentage contribution

is generally low for a particular variable, then this variable does not contribute to the overall low or high predictability of $O_2$.

The largest contribution to the normalized variance in $O_2$ at the surface stems from $O_2^{sol}$ (Figure 7) with a globally averaged contribution of 58%, followed by AOU with 23% and the covariance between $O_2^{sol}$ and AOU contributing 19%. Thus, the $O_2^{sol}$ predictability pattern (Figure 7b) is almost identical to the $O_2$ predictability pattern (Figure 7a or Figure 3c), i.e. high

predictability in the North Atlantic, Southern Ocean and the Arctic, and low predictability in the mid-latitudes. As $O_2^{sol}$ at the ocean surface is mainly controlled by temperature (Garcia and Gordon, 1992), it is not surprising that the pattern of surface $O_2$ predictability (Figure 7a and 3c) is also almost identical to the pattern of SST predictability (Figure 3a). In the Arctic Ocean and around Antarctica, however, AOU (Figure 7f) is almost solely responsible for the normalized variance of $O_2$. As a result, the predictability time horizon of $O_2$ (Figure 7a) is similar to the AOU predictability time horizon (Figure 7c) in these two

regions. The covariance between $O_2^{sol}$ and AOU overall plays a minor role (Figure 7g).

The picture is quite different at 300-m depth (Figure 8), where the largest contribution percentage-wise to the normalized variance of $O_2$ stems from AOU (64% on global average), with minor contributions from $O_2^{sol}$ (13%) and the covariance between $O_2^{sol}$ and AOU (23%). Therefore, the pattern of the AOU predictability time horizon (Figure 8c) is similar to the

pattern of the $O_2$ predictability time horizon (Figure 8a). Exceptions are found in the eastern equatorial Pacific, where the covariance dominates (Figure 8g) and the northern North Atlantic, where $O_2^{sol}$ dominates (Figure 8e). The dominance of AOU in explaining subsurface $O_2$ predictability is also the reason why $O_2$ predictability generally increases with depth (Figure 5c), which is not the case for temperature (Figure 5a).



### 3.3.3 pH

The predictability characteristics of pH are decomposed into its primary drivers in the marine carbonate system, namely temperature, salinity, DIC and Alk (Figure 9). Even though the total normalized ensemble variances from the Taylor deconvolution are only approximations of the total real ensemble variances due to nonlinearities in carbonate chemistry, the values of the Taylor deconvolution are always within ±2% of the real values giving us confidence in the appropriateness of the Taylor deconvolution method for pH.


   At the surface, the largest contribution percentage-wise stems from the covariance between Alk and DIC (Figure 9j; with 26% globally averaged), followed by DIC (Figure 9i; 22%), Alk (Figure 9h; 15%), the covariance between SST and DIC (Figure 9k; 14%), and SST (Figure 9g; 9%). All other possible contributors such as sea surface salinity and its covariances (including the covariance between SST and Alk) are not discussed further, as their contributions are below 5%. The pH predictability at

the surface is therefore mainly determined by Alk and DIC, and to a lesser extent SST. The high potential predictability of pH in the North Atlantic, the Arctic Ocean and in the eastern North Pacific, and the low predictability in the tropical regions (Figure 9a and Figure 3c) are mainly determined by DIC and Alk and the covariance between DIC and ALK. SST plays a role for parts of the North Atlantic. The predictability of pH in the Southern Ocean is mainly determined by DIC, SST and their covariance. Even though SST exhibits enhanced predictability in the Southern Ocean in relation to pH, the low predictability

of DIC and the covariance of DIC and SST leads to the overall diminished predictability time horizon for pH relative to SST there.

   The pH predictability at 300-m depth (Fig. 10a) is mainly determined by DIC (accounts for 44% on global scale; Fig. 10j), and to a lesser extent by the covariance between DIC and SST (19%; Fig. 10k) and the covariance between Alk and DIC (15%;

Fig. 10j). Interestingly, the relatively low pH predictability of about 5 yr in the western equatorial Pacific and the northern Indian Ocean is also mainly determined by DIC (Fig. 10d,i) and the covariance between DIC and SST (Fig. 10f,k). The low predictability of pH in the South Pacific is caused by the covariance between SST and DIC. Again, salinity plays a negligible role (not shown).

### 3.3.4 Net primary production

To understand the drivers that may set the upper limits of NPP predictability, we first split the NPP into the contributions from small phytoplankton production ($NPP_{sm}$), large phytoplankton production ($NPP_{Lg}$) and production by diazotrophs ($NPP_{Di}$; see section 2.3.2 and Appendix A). The largest contribution (i.e. the most important driver of NPP potential predictability) stems from $NPP_{sm}$ (65% averaged globally; Figure 11). The second most important contributor is the covariance between $NPP_{sm}$ and $NPP_{Lg}$ (19%) followed by $NPP_{Lg}$ (9%). Diazotrophs and all other covariances have only a small impact on the predictability

of NPP (< 5%; not shown in Figure 11). The large dominance of $NPP_{sm}$ is not unexpected as the small phytoplankton production



overall dominates the total phytoplankton production in ESM2M (Dunne et al., 2013; Laufkötter et al., 2015). $NPP_{sm}$ accounts for 84% of the total NPP at global scales, whereas $NPP_{Lg}$ and $NPP_{Di}$ only account for 14% and 2%, respectively.

On regional scales, $NPP_{sm}$ determines almost everywhere the predictability of NPP (Figure 11f). Exceptions are the eastern
equatorial Pacific and the higher northern latitudes, where $NPP_{Lg}$ (Figure 11e) and the covariance between $NPP_{Lg}$ and $NPP_{sm}$ (Figure 11g) also play a substantial role. Interestingly, the $NPP_{Lg}$ (Figure 11b) has overall greater predictability than NPP (Figure 11a) and $NPP_{Sm}$ (Figure 11c).

To understand the drivers of small phytoplankton predictability, we further deconvolve $NPP_{Sm}$ into growth rate and small
phytoplankton stock (Figure 12; equation 6 in section 2.3.2). The deconvolution suggests that the largest contribution to the potential predictability on a global scale stems from the small phytoplankton stock (51%) followed by the growth rate (31%) and the covariance between stock and growth rate (18%). Between 40°S and 40°N, the $NPP_{Sm}$ predictability is almost solely determined by the small phytoplankton stock, with the exception of the eastern equatorial Pacific, where the growth rate is more important. Also, the low $NPP_{Sm}$ predictability in the North Atlantic mainly originates from the variance of the stock,
indicated by the lower predictability of the stock compared to the growth rate there. As we stated previously, NPP has a relatively low potential predictability over the Southern Ocean compared to the other ecosystem drivers. Our analysis shows that small phytoplankton (Figure 11) and especially the growth rate of the small phytoplankton (Figure 12) is important for setting this local minimum (Figure 10).

We further deconvolute the drivers of the surface growth rate predictability of small phytoplankton into its temperature, nutrient and light limiting factors (see Eq. 7 in section 2.3.2; Figure 13). As the limiting factors are not saved routinely as 3-dimensional fields, we focus here on the growth rate and its limiting factors at the surface. Note that the growth rate predictability at the surface (Figure 13a) may differ from the growth rate predictability integrated over the top 100 m (Figure 12c), especially in the Southern Ocean and the North Atlantic. At the surface and at the global scale, the largest contribution
stems from the nutrient limitation term (50%) followed by the temperature limitation term (25%) and the covariance between the temperature and nutrient limitations (13%). At the regional scale, the nutrient limitation term clearly dominates at mid-latitudes (Figure 13). This is in contrast to the higher latitudes and the eastern equatorial Pacific, where the temperature limitation term is dominant. The light limitation term only plays a substantial role (up to 20%) around Antarctica and close the Arctic sea ice edge. The observed high predictability for NPP in the mid-latitudes can therefore be attributed to the high
predictability of the nutrient limitation, especially given that the growth rate predictability at surface is similar to the growth rate predictability integrated over the top 100 m in this region. At latitudes north of 40°N and south of 40°S, the temperature limitation is the most important contributor. Therefore, the predictability pattern of the growth rate strongly resembles the one for SST in these regions. In the Southern Ocean, however, the growth rate predictability at surface is much larger than the



growth rate predictability integrated over top 100 m indicating that a process different than temperature (e.g. light limitation)
may limit predictability there.

**4 Discussion and Conclusion**

We set out three goals for this study: (a) assessing the global characteristics of potential predictability for temperature, pH, $O_2$
and NPP, as a mean to identify an upper bound on our ability to predict conditions for marine ecosystems, (b) assessing regional
and depth-dependent characteristics of potential predictability, and (c) identifying the potential mechanisms that limit or
increase predictability for the different marine ecosystem drivers. This was pursued within a perfect modelling framework
using a comprehensive Earth system model.

The analysis revealed that on global scales the potential predictability is surprisingly similar, i.e. three years for all four marine
ecosystem drivers (Figure 2; first goal), despite the fact that the regional processes operating are different over a range of scales
(second and third goal). This is unexpected, as the ocean processes that sustain the disparate divers should not be expected to
have identical memory as pertains to predictability. For example the relatively high potential predictability identified for SST
and surface $O_2$ over the subpolar North Atlantic (the SST to be consistent with Griffies and Bryan 1997; Boer 2000; Collins et
al. 2006; Keenlyside et al. 2008) and the Southern Ocean (consistent with Zhang et al. (2017) and Marchi et al. (2019)) is not
reflected in NPP. Likewise, the high potential predictability of NPP in the subtropical gyres is not simulated for other ecosystem
drivers and the low predictability of surface pH in the Southern Ocean is reflected in neither SST nor in surface $O_2$.

Our results suggesting the same global predictability horizon for all four ecosystem drivers is not inconsistent with time of
emergence diagnostics for transient climate warming scenarios where pH (early emergence) and NPP (late emergence) behave
opposite (Frölicher et al., 2016; Rodgers et al., 2015; Schlunegger et al., 2019). Time of emergence is defined as the ratio
(large for pH and small for NPP) of the anthropogenic forced change to the background internal variability. Comparing our
results with the time of emergence analysis is therefore complicated by the presence of the anthropogenic forced signal in
scenario projections. In fact it is the presence of the large invasion flux for $CO_2$ that renders acidification the most rapidly
emergent of the drivers under anthropogenic perturbations, in particular relative to NPP. The similarities between the analyses
of predictability and emergence timescales lie in the noise, which is expected to include not only modes of climate variability
such as ENSO, but also higher frequency variability such as cloud cover that may impact NPP for both cases.

Our study complements earlier studies which suggested that marine ecosystem drivers may be predictable on multi-annual
timescales. In contrast to earlier studies (Chikamoto et al., 2015; Park et al., 2019; Séférian et al., 2014b), rather than focusing
on a single ecosystem driver, we compare and contrast the potential predictability of four marine ecosystem drivers and also
evaluate the processes behind their respective predictability limits. We find that in contrast to SST, these ecosystem drivers



depend on a complex interplay between physical and biogeochemical underlying processes. For $O_2$, the importance of subsurface AOU reveals a complex interplay between non-local circulation and biological consumption, whereas at the surface, $O_2$ is mainly determined by the predictability of SST. For NPP, the growth rate of the small phytoplankton in the Southern Ocean is important for setting the local minimum in predictability there. The potential predictability of surface pH is mainly

determined by a complex interplay between DIC and Alk predictability in the low latitudes and DIC, Alk and temperature predictability in high latitudes. Interestingly, we find higher potential predictability for SST than for NPP in the equatorial Pacific, which is in contrast to findings of Séférian et al. (2014a). Importantly, this may be indicative of a potential model-dependency of the relationship between ecosystem driver predictability. Séférian et al. (2014b) attributed larger NPP predictability to the idea that the nutrient supply processes that modulate NPP are themselves regulated by thermocline wave

adjustment processes, without sizeable modulation by surface fluxes. This was framed as standing in contrast to the case of SST, where air-sea fluxes reflecting higher-frequency variations act to reduce the predictability of SST. In ESM2M, the predictability for SST in the eastern equatorial Pacific (biome 6 in Figure 4) is approximately 3.5 yr, modestly longer than the predictability for NPP of approximately 3 yr. In ESM2M, NPP is only weakly correlated with changes in upwelling and nutrient supply in the eastern tropical Pacific (as was shown in Figure 2 of Kwiatkowski et al. (2017)). This is confirmed by our analysis

showing that nutrient limitation is not the dominant term for explaining the predictability of NPP there. This indicates that less predictable processes occurring over shorter timescales, such as temperature and/or light level variations, influence NPP predictability.

Even though we consider our conclusion as robust, a number of potential caveats warrant discussion. These include the (i)

ensemble design of the perfect model simulations (e.g. initialization and number of ensemble members) and (ii) the impact of model biases. First, our simulations are all initialized with SST perturbations applied to a single grid cell in the Weddell Sea and therefore a different spatial perturbation strategy may give different results. However, as the signal at the ocean surface spreads very rapidly, i.e. after four days all grid cells at the ocean surface are perturbed, our results are insensitive to the spatial initialization method, at least in the upper ocean. Second, all ensemble simulations start in January 1$^{st}$ of the corresponding

simulation year. It has been shown that the forecast skill of seasonal predictions may depend strongly on the way the models are initialized. ENSO forecasts, for example, have a much lower predictability if they are initialized before and through spring (Webster and Yang, 1992). However, as our focus is on annual-to-decadal timescales, this effect is less important for our analysis. Third, we have employed only six starting points for our 40-member ensemble simulation. Even though all six ensemble simulations branched off at different El Niño Southern Oscillation states of the preindustrial control simulation, our

choice of six macroperturbations may still introduce aliasing issues that could bias our results. Although the computing resources at our disposal for this study did not allow for expanding the number of starting points, we recommend that future studies with CMIP-class models should expand the number of initialization points to further explore the sensitivity of the results to the starting point of the ensembles. The second caveat in our study is that we only used one single Earth system model and that our results might depend on the model formulation and resolution. Even though the GFDL ESM2M model



achieves sufficiently fidelity in its preindustrial states (Bopp et al., 2013; Dunne et al., 2012, 2013; Laufkötter et al., 2015), it is well known that CMIP5-generation models have imperfect representation of variability over a range of timescales, ranging from weather variability to ENSO variability (Frölicher et al., 2016; Resplandy et al., 2015). It would be necessary to repeat our predictability experiments with a set of different Earth system models to investigate the dependence of our result on the model representation (Séférian et al., 2018), in parallel with broader efforts to further evaluate noise characteristics of these

models. Also, the ocean model resolution of GFDL ESM2M is rather coarse and cannot represent the critical scales of small-scale structures of circulation. Predictability studies using high resolution ocean models with improved process representations are therefore needed to explore potential predictability, especially at the local scale. However, it is currently impossible in many cases to constrain the simulated variability in biogeochemical drivers, especially for the ocean subsurface, with observations due to limited data availability (Frölicher et al., 2016; Laufkötter et al., 2015).


Currently, no global coupled physical-biogeochemical seasonal-to-decadal forecast system is yet operational (Tommasi et al., 2017). However, our study suggests great promise that physical-biogeochemical forecast systems may have the potential to provide useful information to a wide group of stakeholders, such as, for example, for the management of fisheries (Dunn et al., 2016; Park et al., 2019). Our study therefore underscores the need to further develop integrated physical-biogeochemical

forecast systems. Especially in regions with high predictability, such as the North Atlantic (for temperature, $O_2$, pH), the Southern Ocean (for temperature and $O_2$), and mid-latitudes (for NPP), installing and maintaining a spatially and temporally dense physical and biogeochemical ocean observing system would have the potential to significantly improve the effective predictability of marine ecosystem drivers.

**Appendix A**

The NPP in TOPAZ2, defined as the phytoplankton nitrogen production, is individually described for all phytoplankton groups $i$ by the product of a phytoplankton growth rate $\mu_i$ and the amount of nitrogen in the plankton group $[N]_i$:

$$NPP_i = \mu_i \cdot [N]_i. \tag{A1}$$

The growth rate of the small phytoplankton group is given by a maximum growth rate times the limiting factors of nutrients $N_{lim}$, light $L_{lim}$, and temperature $T_f$:

$$\mu_{Sm} = \frac{\mu_{max'}}{1+\zeta} \cdot N_{lim} \cdot L_{lim} \cdot T_f. \tag{A2}$$

The temperature limitation factor is:



$$T_f = \exp(k_{epp} \cdot T). \tag{A3}$$

The nutrient limitation factor is:


$$N_{lim} = \min(N_{\text{Fe}}, N_{\text{PO}_4}, N_{\text{NO}_3} + N_{\text{NH}_4}), \tag{A4}$$

with iron limitation:

$\quad N_{\text{Fe}} = \frac{Q_{\text{Fe:N}}^2}{Q_{\text{Fe:N}}^2 + K_{\text{Fe:N}}^2}, \quad$ with $\quad Q_{\text{Fe:N}} = \min\left(Q_{\text{Fe:N max}}, \frac{[\text{Fe}]_{Sm}}{[\text{N}]_{Sm}}\right),$ $\tag{A5}$

with phosphate limitation:

$$N_{\text{PO}_4} = \frac{Q_{\text{P:N}}}{Q_{\text{P:N max}}}, \quad \text{with} \quad Q_{\text{P:N}} = \min\left(Q_{\text{P:N max}}, \frac{[\text{P}]_{Sm}}{[\text{N}]_{Sm}}\right), \tag{A6}$$


with nitrate limitation:

$$N_{\text{NO}_3} = \frac{[\text{NO}_3]}{[\text{NO}_3] + K_{\text{NO}_3}} \cdot \frac{1 + [\text{NH}_4]}{K_{\text{NH}_4}}, \tag{A7}$$

and with ammonium limitation:

$$N_{\text{NH}_4} = \frac{[\text{NH}_4]}{[\text{NH}_4] + K_{\text{NH}_4}}. \tag{A8}$$

The light limitation factor is:

$\quad L_{lim} = 1 - exp\left(\frac{-\alpha\theta[IRR]}{N_{lim}T_f\mu_{max}}\right),$ $\tag{A9}$

with $\quad \theta = \frac{\theta_{max} - \theta_{min}}{1 + (\theta_{max} - \theta_{min})\alpha[IRR_{mem}]/(2N_{lim}T_f\mu_{max})} + \theta_{min},$ $\tag{A10}$

and $\quad \theta_{min} = \max\left(0, \theta_{min}^{nolim} - \theta_{min}^{lim}\right) \cdot N_{lim} + \theta_{min}^{lim},$ $\tag{A11}$


where $[IRR]$ describes the photosynthetically active radiation and $[IRR_{mem}]$ is the irradiation memory over the last 24 hours.





**Appendix B**

Potential predictability may depend on the choice of the predictability metric (Hawkins et al., 2016). Therefore, we calculate two additional metrics to assess the robustness of our results: the normalized root mean square error (NRMSE) and the intra-

ensemble anomaly correlation coefficient (ACC$_I$). The NRMSE is similar to the PPP but uses standard deviations instead of variances and compares every ensemble member to every other member of that ensemble, thereby increasing the effective sample size (Collins et al., 2006):

$$\text{NRMSE}(t) = 1 - \sqrt{\frac{\langle (X_{ij}(t) - X_{kj}(t))^2 \rangle_{i,j,k \neq i}}{2\sigma_c^2}} \qquad \text{(B1)}$$


$\langle \cdot \rangle$ means that we sum over the listed indices and divide by the degrees of freedom. The intra-ensemble anomaly correlation coefficient (ACC$_I$) is a measure for the correlation between the anomaly of all ensemble members of an ensemble averaged over all ensembles and is regularly used for assessing operational predictions (Goddard et al., 2013). The anomaly is defined as the deviation of a given value from the climatological mean $\mu_j$ (i.e. the mean over the control run) over the $j$-th ensemble

period.

$$\text{ACC}_I(t) = \frac{\langle (X_{ij}(t) - \mu_j)(X_{kj}(t) - \mu_j) \rangle_{i,j,k \neq i}}{\langle (X_{ij}(t) - \mu_j)^2 \rangle_{i,j}} \qquad \text{(B2)}$$

While PPP and NRMSE estimate predictability by comparing the spread of the ensembles to the natural variability from the

control simulation, the anomaly correlation coefficients include the phase alignment of the ensembles and the control simulation. We again use a F-test for NRMSE and a t-test for ACC$_I$ to estimate the predictability threshold.

Figure B1 compares the two additional metrics applied to SST with the PPP metric. We introduce an artificially predictability threshold for ACC$_I$ in such a way that the emerging pattern matches the PPP predictability time horizon best. This allows us

to compare the relative differences in predictability between the metrics best. The predictability pattern for SST obtained from all three metrics are very similar. Especially the patterns obtained using PPP and NRMSE are nearly identical. This can be expected since both the PPP and the NMRSE estimate potential predictability by analyzing the ensemble spread. The ACC$_I$ shows some small differences to PPP and NMRSE, especially in the Southern Ocean and the North Pacific.

**Data availability**

The GFDL ESM2M simulations are available upon request.



**Author contributions**

TLF, KBR, LR, and CCR designed the study. TLF set up the ensemble simulations and KBR performed the simulations. LR performed most of the analysis. TLF wrote the initial manuscript. All authors contributed significantly to the writing of the paper.

**540**   **Competing interests**

All authors declare no competing interests

**Acknowledgements**

TLF and LR acknowledges support from the Swiss National Science Foundation under grant PP00P2_170687, from the European Union's Horizon 2020 research and innovative programme under grant agreement No 821003 (CCiCC) and from
**545**   the Swiss National Supercomputing Centre (CSCS). Support for KBR was provided by the Institute for Basic Science project code IBS-R028-D1. The authors thank Friedrich Burger for discussions on the Taylor deconvolution method.

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

**Figures**

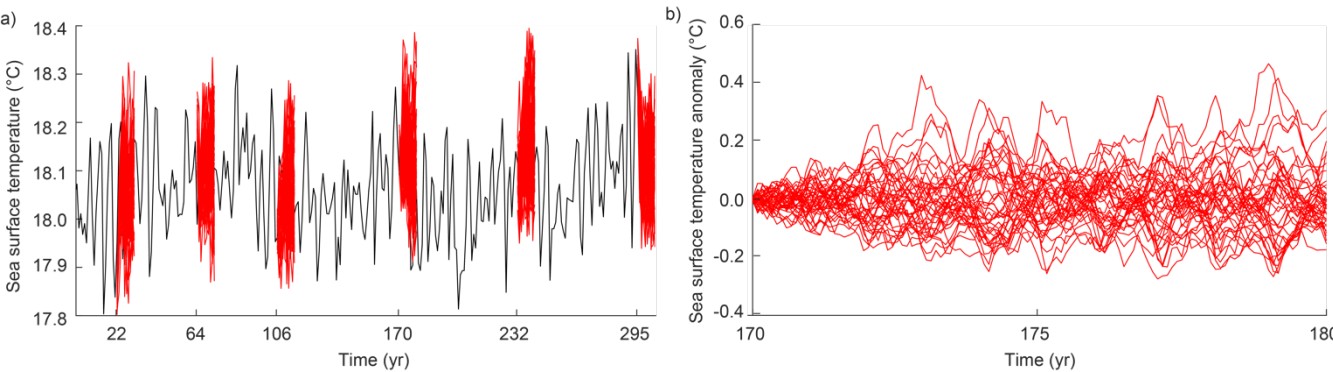


**Figure 1: Illustration of the model setup.** (a) Simulated global mean SST of the 300-yr reference control simulation (black line) and of the

six 10-yr long 40 ensemble simulations (red lines). (b) Global mean SST anomaly (i.e., deviation from the control simulation) for the first

ensemble simulation starting in year 170.






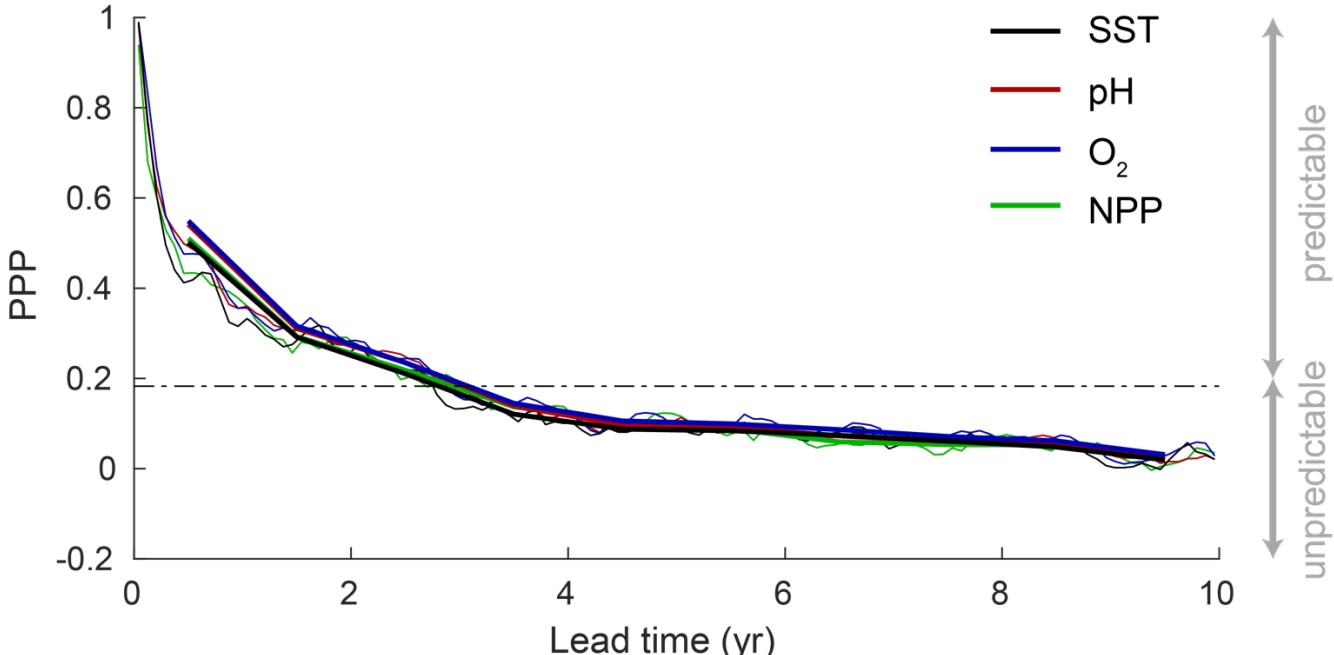

**Figure 2: Globally averaged prognostic potential predictability (PPP) for all four marine ecosystem drivers at the surface, except for NPP which is integrated over the top 100 m.** Monthly mean (thin lines) and annual mean (thick lines) values of PPP are shown. The horizontal black dashed line represents the predictability threshold. If PPP is above (below) the predictability threshold, the driver is potentially predictable (unpredictable) as indicated with the arrows on the right hand side. The PPP has first been calculated at each grid cell

and then averaged globally.





**Figure 3: Predictability time horizon for (a) SST, (b) surface pH, (c) surface O₂, and (d) NPP integrated over top 100 m using PPP as predictability measure.** The contour lines in (d) indicate the annual mean total nitrogen production in mol N kg⁻¹ yr⁻¹ averaged over the 300-yr preindustrial control simulation to highlight regions with low and high NPP. In (d) regions north of 69°N and south of 69°S have been excluded since NPP is zero during winter time there.






**Figure 4: PPP for all four ecosystem drivers averaged over 17 different biomes at the surface, except for NPP, which is integrated over top 100 m.** Monthly means are shown as thin lines and annual means as thick lines. The horizontal dashed black lines in each panel represents the predictability threshold. The lower right panel shows the boundaries and the geographical location of the biomes 1 to 17.


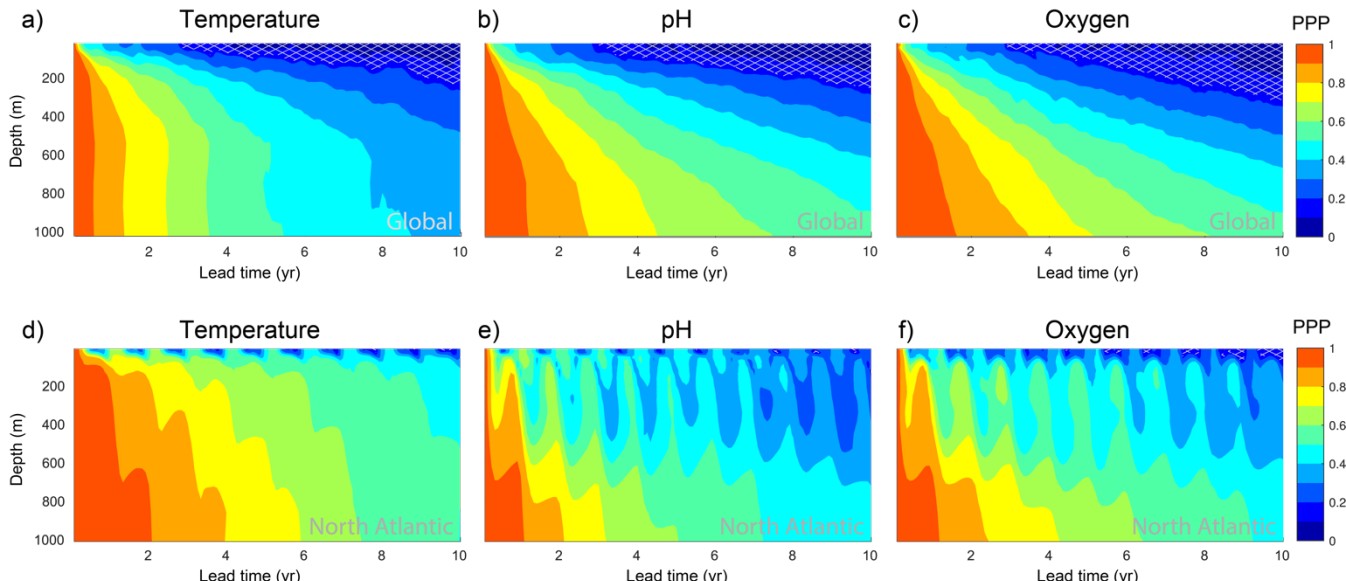

**Figure 5: PPP depth profiles for the top 1000 m for ocean temperature, oxygen and pH at the (a-c) global scale and (d-f) in the North Atlantic.** The PPP is shown as monthly means. The light gray hatching indicates a PPP value below the predictability threshold. The North Atlantic is defined as the ocean area between 40°N and 60°N in the North Atlantic. Note that the variance over the control simulation for pH

is zero for approximately 0.4% of grid cells at subsurface, which leads to an undefined PPP value there (see Eq. 2). Such grid cells have been excluded here.





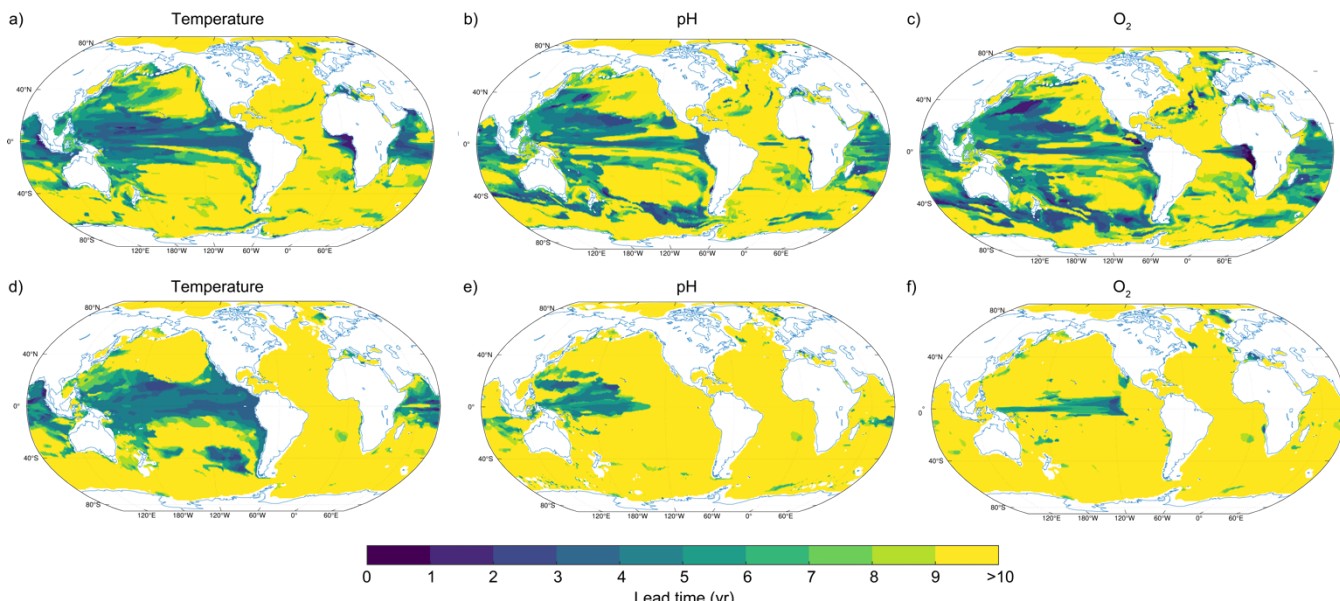

**Figure 6: Spatial pattern of the predictability time horizon at (a-c) 300-m and (d-f) 1000-m depth for (a,d) ocean temperature, (b,e) pH, and (c,f) dissolved oxygen.**







**Figure 7: Spatial pattern of the (a-d) predictability time horizons and (e-g) contribution of different terms to the predictability of**

**oxygen at the surface.** (a-d) Predictability time horizon for (a) $O_2$, (b) $O_2^{sol}$, (c) AOU, and (d) covariance between $O_2^{sol}$ and AOU. (e-g)

Percentage contributions of (e) $O_2^{sol}$, (f) AOU and (g) covariance between $O_2^{sol}$ and AOU relative to the sum of all terms. Red shading in (e-

g) represents positive absolute values of the variance and covariance terms. The percentage contributions are shown as averages over the

entire 10 yr of the simulations. The percentage contributions do not change substantially over the 10 yr (always within ± 5% of the 10-yr

averages).




**Figure 8: Same as Figure 7, but at 300-m depth.**







**Figure 9: Spatial pattern of the (a-f) predictability horizons and (g-k) contribution of different terms to the predictability of pH at the surface.** (a-f) Predictability time horizon for (a) pH, (b) SST, (c) Alk, (d) DIC, and the covariance between (e) Alk and DIC, and (f) DIC and SST. (g-k) Percentage contributions of (g) SST, (h) Alk, (i) DIC, and covariance of (j) ALK and DIC, and (k) DIC and SST relative to the sum of all terms. Red shading in (g-k) represents positive absolute values of the variance and covariance terms. The percentage contributions are shown as averages over the entire 10 yr of the simulations. The percentage contributions do not change substantially over

the 10 yr (always within ± 5% of the 10-yr averages). Note that the terms that do not contribute to pH predictability such as sea surface salinity, and the covariances between sea surface salinity and all other terms as well as the covariance between SST and Alk are not shown here.







**Figure 10: Same as Figure 9, but at 300-m depth.**

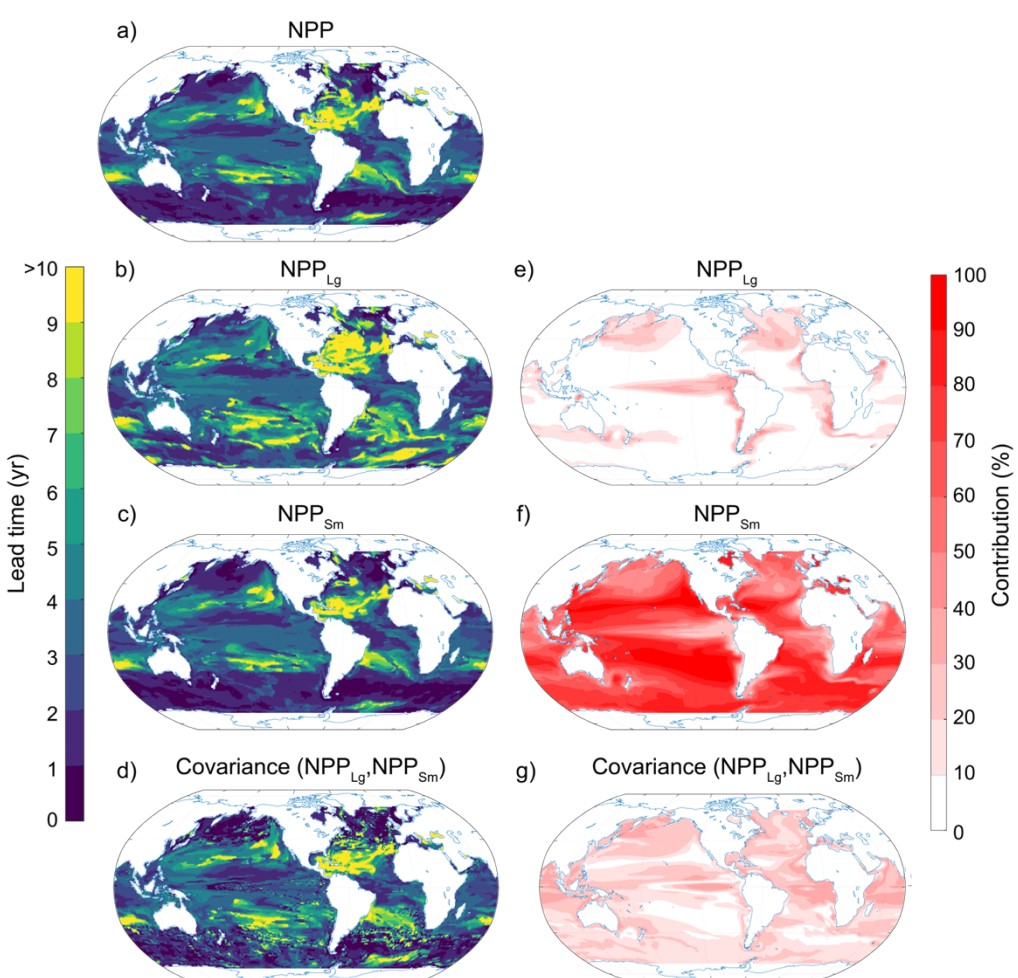

**Figure 11: Spatial pattern of the (a-d) predictability horizons and (e-g) contribution of different terms to the predictability of NPP**

**integrated over the top 100 m.** (a-d) Predictability time horizon for (a) NPP, (b) large phytoplankton production $NPP_{Lg}$, (c) small

phytoplankton production $NPP_{Sm}$, and (d) the covariance between $NPP_{Lg}$ and $NPP_{Sm}$. (e-g) Percentage contributions of (e) $NPP_{Lg}$, (f) $NPP_{Sm}$,

(g) and covariance of $NPP_{Lg}$ and $NPP_{Sm}$ relative to the sum of all terms. Red shading in (e-g) represents positive absolute values of the

variance and covariance terms. The percentage contributions are shown as averages over the entire 10 yr of the simulations. The percentage

contributions do not change substantially over the 10 yr (always within ± 5% of the 10-yr averages). Note that the terms that do not

substantially contribute to NPP predictability such diazotrophs ($NPP_{Di}$), and the covariances between $NPP_{Di}$ and all other terms are not

shown here.





**Figure 12: Spatial pattern of the (a-d) predictability horizons and (e-g) contribution of different terms to the predictability of small phytoplankton production (NPP$_{Sm}$) integrated over the top 100 m.** (a-d) Predictability time horizon for (a) NPP$_{Sm}$, (b) small phytoplankton stock, (c) growth rate of small phytoplankton, and (d) the covariance between the stock and the growth rate of small phytoplankton. (e-g) Percentage contributions of (e) stock, (f) growth rate, (g) and covariance of stock and growth rate relative to the sum of all terms. Red shading in (e-g) represents positive absolute values of the variance and covariance terms. The percentage contributions are shown as averages over the entire 10 yr of the simulations. The percentage contributions do not change substantially over the 10 yr (always within ± 5% of the 10-yr averages).





**Figure 13: Spatial pattern of the (a-e) predictability horizons and (f-i) contribution of different terms to the predictability of the small phytoplankton growth at the surface.** (a-f) Predictability time horizon for (a) growth rate of small phytoplankton, (b) temperature limitation, (c) nutrient limitation, (d) light limitation, and (e) the covariance between the temperature and nutrient limitation. (f-i) Percentage contributions of (f) temperature limitation, (g) nutrient limitation, (h) light limitation, and (i) covariance between temperature and nutrient

790   limitation relative to the sum of all terms. Red shading in (f-i) represents positive absolute values of the variance and covariance terms. The percentage contributions are shown as averages over the entire 10 yr of the simulations. The percentage contributions do not change substantially over the 10 yr (always within ± 5% of the 10-yr averages). Note that the terms that do not substantially contribute to NPP predictability covariances between temperature and light and nutrient are not shown here.

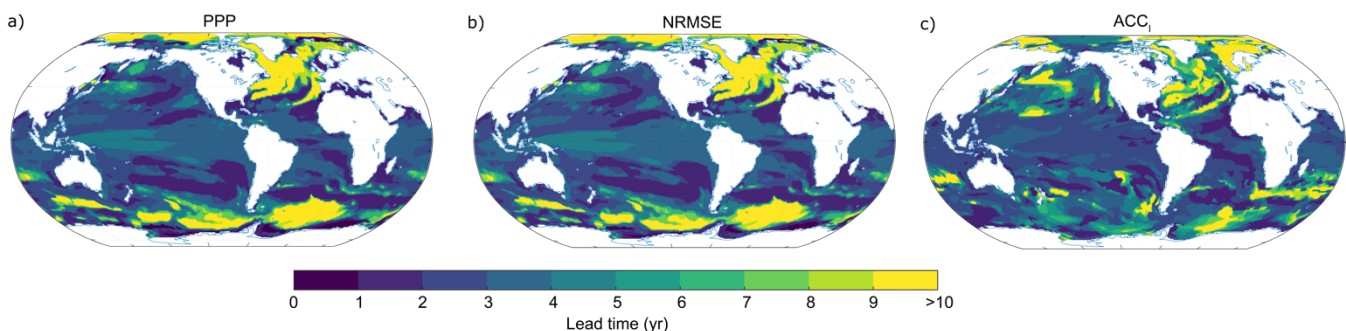

795

**Figure A1: SST predictability time horizon calculated with different metrics.** Spatial pattern of the predictability horizon for sea surface temperature using (a) PPP, (b) NRMSE, and (c) ACC$_I$. Note that we assume an arbitrary predictability threshold for ACC$_I$ so that the emerging pattern matches the PPP predictability best. This allows us to compare the relative differences in predictability.




800   **Table A1: TOPAZ2 parameters for small phytoplankton**

| Parameter | Value | Units | Description |
| --- | --- | --- | --- |
| $\zeta$ | 0.1 | | Photorespiration loss |
| $k_{epp}$ | 0.063 | $C^{-1}$ | Temperature coefficient for growth |
| $\alpha$ | 2.4e-5 · 2.77e18/6.022e17 | g C g Chl$^{-1}$ m$^2$ W$^{-1}$ s$^{-1}$ | Light harvest coefficient |
| $\mu_{max}'$ | 1.5e-5 | s$^{-1}$ | Maximum growth rate at 0°C |
| $\theta_{min}^{nolim}$ | 0.01 | g Chl g C$^{-1}$ | Minimum Chl:C without nutrient limitation |
| $\theta_{min}^{lim}$ | 0.001 | g Chl g C$^{-1}$ | Minimum Chl:C with complete nutrient limitation |
| $\theta_{max}$ | 0.04 | g Chl g C$^{-1}$ | Maximum Chl:C |
| $K_{NO_3}$ | 2e-6 | mol N kg$^{-1}$ | NO$_3$ half-saturation coefficient |
| $K_{NH_4}$ | 2e-7 | mol N kg$^{-1}$ | NH$_4$ half-saturation coefficient |
| $K_{Fe:N}$ | 12e-6 · 106/16 | mol Fe mol N$^{-1}$ | Half-saturation coefficient of iron deficiency |
| $Q_{Fe:N\,max}$ | 46e-6 · 106/16 | mol Fe mol N$^{-1}$ | Maximum Fe:N limit |
| $Q_{P:N\,max}$ | 0.1458 | mol P mol N$^{-1}$ | Maximum P:N limit |