# Peer review of "Potential predictability of marine ecosystem drivers"

_Biogeosciences, 2019_

## Referee Comment (RC1) · Anonymous Referee #1 · 31 Jan 2020

This paper evaluates the potential predictability of marine ecosystem drivers (T, pH, O2, and NPP) and discusses which physical and biological processes fundamentally determines the upper limit of the predictability at global and regional scales using a series of perfect model simulation in an earth system model (GFDL ESM2M). Whereas previous studies focus on a physical process that describes the potential predictability of marine biogeochemical variables, this study shows that the biogeochemical interactions, as well as physical processes, are key drivers to contribute to the upper limit of the predictability, which leads to the understanding of differences in the potential predictability between temperature (physical) and biogeochemical variations. The global and regional characteristics of the biogeochemical predictability that this paper shows are important information to produce skillful multi-year predictions of marine ecosys-

tems. I recommend a minor revision according to the comments given below.

(1) To what extent does AMOC variability affect the potential predictability of marine ecosystem drivers in the North Atlantic?

The decadal potential predictability of marine ecosystem drivers is prominent in the North Atlantic, which would be strongly related to the variability in AMOC. To understand the mean states and variations in AMOC in perfect and ensemble simulations in GFDL ESM2M, I request to show the time series of AMOC variability, like Figure 1a.

(2) What limiting nutrient contributes to higher potential predictability in NPP?

In this paper, the regions of higher potential predictability in NPP (Fig. 3d) correspond to those of higher predictability in nutrient limitation, which suggests that higher potential predictability in NPP is fundamentally constrained by the availability of limiting nutrients. Since the model uses the formulations of limitation by multiple nutrients, I wonder what nutrient is key to contribute to higher potential predictability in NPP at the regional scale. The figure and description of a spatial pattern of limiting nutrients are helpful to understand the characteristics of long-term biological variability.

Other comments:

L137. Remove "The prognostic "

L159. reflect, not reflects

L. 281. largely, instead of large

L 363. Why do you refer to Fig.10?

L. 527. Figure B1 should be Figure A1 on page 37.

Page 800. The unit of the iron half-saturation coefficient would be wrong.

Figure 3d, Contour lines, and contour number information are too dark to be identified.

[Figure]

---

## Referee Comment (RC2) · M. Baird (Referee) · 1 Mar 2020

Journal: Biogeosciences (BG) Title: Potential predictability of marine ecosystem drivers Author(s): Thomas L. Frölicher, Luca Ramseyer, Christoph C. Raible, Keith B. Rodgers, and John Dunne MS No.: bg-2019-506 MS Type: Research article

This manuscript is a well written, interesting analysis of limits to prediction of biogeochemical quantities in a global Earth system model. I particularly liked the oceanographic explanations for the outcomes of the numerical experiments, and deconvolution of drivers. I am not an expert in global model predictability, but I suspect it is important for the field to undertake a number of these experiments on different models (the authors say this themselves). Even if similar studies exist, or follow this one,

this study will remain important. Therefore, I recommend publication with the following, relatively easily-addressed points considered:

Major comments:

1. As someone not familiar with this form of model sensitivity analysis, I found some jargon that could have been avoided, or better explained. The term 'perfect modelling framework' was introduced as though the reader should know what it means. More simply the study is a test of the sensitivity of biogeochemical quantities to temperature initial conditions.

2. By using temperature the emphasis is on limits to prediction of physical driving of biogeochemical quantities. Other initial conditions could have equally been perturbed, such as salinity, nutrients etc. And of course there are many other factors limiting predictability, such as the model parameterisation. This isolation of one source of limits to prediction is appropriate but should be made clearer in the introduction, and then discussed, in the light of the results, more thoroughly in the Discussion. For example, changing BGC models, or even the remineralisation rate of organic matter, would change the time scale of AOU.

3. [Most important point that needs addressing]. The terms "lead time" and "predictability time horizon" are used interchangeably in the last paragraph of p8, which demonstrates an inconsistency. The term lead time makes sense to me in Fig 2, 4 and 5. It is the time axis, starting at the perturbed time, along which the variability of the ensemble and controls are measured. But figures 3, 6, 7 etc. the surface plotted is labelled "lead time" when it should be "predictability time horizon". Predictability time horizon is loosely describes as PPP < 0.183 (also, is prognostic potential predictability the same thing as predictability time horizon)? But this is problematic since PPP varies with time. Should it be min t for which PPP(t) < 0.183? Check every use of lead time, PPP and predictability time horizon and make sure it is consistent in the manuscript. Predictability is also loosely defined, and would in many cases be best replaced with

"predictability time horizon".

4. Fig. 1b. This figure would be more instructive if it was used as an example of the calculation of the PPP. If the control variance could be plotted (maybe a grey between +/- sigma) and then PPP, and the point at which PPP drops below 0.183. This would set up the rest of the manuscript better. Also, in the caption, why do you call it the "first ensemble simulation".

5. Paragraph 396 – 405 needs re-writing for clarity. I think it is trying to say that you can have different time scales for predictability for perturbations in forcing (such as anthropogenic CO2) to perturbations in initial conditions (as studied here).

6. The discussion has too much focus on obvious limitations (such as ensemble size, years started etc.) and less on more subtle limitations like time-scale of coefficients in the BGC model. The second are particularly worth of discussion here because the effort at deconvolution of the processes allows for an insightful discussion of these.

Minor comments.

1. Whenever referring to time values, try to keep the adjectives to ones with a sense of time such as low -> short (L254), elevated -> lengthened (L195), high - > long (L391). This aids readability. There are many examples of this.

2. The sentence L91 starting "The six "should come before the "Note" for better read-ability.

3. L92 replace "are" with "were"?

4. Line 100. For those interpreting the equation, maybe a sentence after it "Thus the range of perturbations is evenly spread from -0.002 to 0.002 C with the control in the centre."

5. L110 replace "underrepresented" with "underestimated"

6. Description of Eq.2 (L119-120) doesn't mention the six ensembles. I didn't fully

understand the rationale for 6 ensembles of size 40. Why not 240 members starting at all different times? Also is sigma of the control the same for all ensembles? Just a little bit more help here to those unfamiliar with the approach.

7. L150 pH is approximately –log10([H+]). I know you didn't mean to define it here, but the use of (or X) sort of implies it.

8. L184 meaning of "PPP with lead time" not clear.

9. L206 replace "across" with "for each of the"

10. L234. How can a coupling enhance predictability? Sentence needs to be more carefully constructed.

11. L387 "predictability of each variable".

---

## Author Comment (AC1) · 10 Mar 2020

**Biogeosciences manuscript bg-2019-506**
"Potential predictability of marine ecosystem drivers" by T. L. Frölicher, et al.
March 10, 2020.

We thank both reviewers for assessing our manuscript. The comments have helped to further improve our manuscript. The review comments are given in black and our reply in blue. Please find attached to the reply a revised manuscript where text changes are highlighted.

**Detailed Response to Reviewer's comments: Referee #1**
This paper evaluates the potential predictability of marine ecosystem drivers (T, pH, O2, and NPP) and discusses which physical and biological processes fundamentally determines the upper limit of the predictability at global and regional scales using a series of perfect model simulation in an earth system model (GFDL ESM2M). Whereas previous studies focus on a physical process that describes the potential predictability of marine biogeochemical variables, this study shows that the biogeochemical interactions, as well as physical processes, are key drivers to contribute to the upper limit of the predictability, which leads to the understanding of differences in the potential predictability between temperature (physical) and biogeochemical variations. The global and regional characteristics of the biogeochemical predictability that this paper shows are important information to produce skillful multi-year predictions of marine ecosystems. I recommend a minor revision according to the comments given below.
We thank the reviewer for this positive and encouraging review.

(1) To what extent does AMOC variability affect the potential predictability of marine ecosystem drivers in the North Atlantic? The decadal potential predictability of marine ecosystem drivers is prominent in the North Atlantic, which would be strongly related to the variability in AMOC. To understand the mean states and variations in AMOC in perfect and ensemble simulations in GFDL ESM2M, I request to show the time series of AMOC variability, like Figure 1a.
We have included a new Figure C1 (see below) that shows the simulated AMOC maximum from the 300-yr long preindustrial control simulation, as well as the prognostic potential predictability of the AMOC maximum. The text in section 3.3.1 has been modified to say: "In GFDL's ESM2M, the AMOC experiences strong low-frequency variability, consistent with Msadek et al. (2010), and its predictability time horizon is about 9 yr (Figure C1)."

[Figure]

Figure C1: (a) Simulated annual mean AMOC maximum of the 300-yr long preindustrial control simulation. The blue line indicates the 10-yr running mean. (b) Monthly mean (thin line) and annual mean (thick line) prognostic potential predictability for the AMOC maximum. The horizontal black dashed line represents the predictability threshold.

Importantly, we note that the predictability of some marine ecosystem drivers is dependent on a complex interplay between physical and biogeochemical processes. NPP, as an illustrative example, has a relatively short predictability time horizon in the North Atlantic, with this behavior not directly prescribed by AMOC variations.

(2) What limiting nutrient contributes to higher potential predictability in NPP? In this paper, the regions of higher potential predictability in NPP (Fig. 3d) correspond to those of higher predictability in nutrient limitation, which suggests that higher potential predictability in NPP is fundamentally constrained by the availability of limiting nutrients. Since the model uses the formulations of limitation by multiple nutrients, I wonder what nutrient is key to contribute to higher potential predictability in NPP at the

regional scale. The figure and description of a spatial pattern of limiting nutrients are helpful to understand the characteristics of long-term biological variability.

Many thanks for this suggestion. We have included in Figure 13f (former Figure 13g) with contours the limiting nutrients (nitrogen, phosphate or iron) for the growth rate of small phytoplankton production at the surface. We added following sentence to section 3.3.4: "In GFDL's ESM2M, the subtropical gyres are mainly iron limited (hatching in Figure 13f) and therefore iron fundamentally constrains the predictability of the growth rate of small phytoplankton there. Exceptions are the boundary region between the subtropical and subpolar gyre in the North Pacific (nitrate limited) as well as the tropical Atlantic (phosphate and nitrate) and the northern Indian Ocean (phosphate)."

However, we would like to stress here that the relatively long predictability horizon for NPP in the subtropical gyres is also strongly influenced by the relatively long predictability horizon of the small phytoplankton stock, as can be seen in Fig. 12e.

Other comments:
L137. Remove "The prognostic "
Done.

L159. reflect, not reflects
Done.

L. 281. largely, instead of large
We have reformulated the sentence to: "Therefore, the influence of high frequency atmospheric variability is large, which leads to diminished potential predictability around Antarctica."

L 363. Why do you refer to Fig.10?
Typo. Many thanks for spotting this. We deleted the reference to Fig. 10, but included a reference to Fig. 3 in the previous sentence.

L. 527. Figure B1 should be Figure A1 on page 37.
We changed the labelling of Figure A1 to Figure B1, as the figure is described in Appendix B.

Page 800. The unit of the iron half-saturation coefficient would be wrong.
Changed and acknowledged.

Figure 3d, Contour lines, and contour number information are too dark to be identified.
We changed the color to red and increased the thickness of the contours.

**Detailed Response to Reviewer's comments: Referee #2**
This manuscript is a well written, interesting analysis of limits to prediction of biogeochemical quantities in a global Earth system model. I particularly liked the oceanographic explanations for the outcomes of the numerical experiments, and deconvolution of drivers. I am not an expert in global model predictability, but I suspect it is important for the field to undertake a number of these experiments on different models (the authors say this themselves). Even if similar studies exist, or follow this one, this study will remain important. Therefore, I recommend publication with the following, relatively easily-addressed points considered.
We thank the reviewer for this positive and encouraging review.

Major comments:
1. As someone not familiar with this form of model sensitivity analysis, I found some jargon that could have been avoided, or better explained. The term 'perfect modelling framework' was introduced as though the reader should know what it means. More simply the study is a test of the sensitivity of biogeochemical quantities to temperature initial conditions.
We clarified in the method section the term 'perfect modelling framework': "By perturbing the initial conditions of the GFDL ESM2M and quantifying the spread of initially nearby model trajectories, the limit of initial condition predictability was assessed. The underlying assumption is that we have a perfect model (e.g. the model accurately represents all physical and biogeochemical processes relevant to assess marine ecosystem drivers at adequate temporal and spatial resolution), near

perfect initial conditions and that we exclude a role for external forcing in determining or limiting predictability."

2. By using temperature the emphasis is on limits to prediction of physical driving of biogeochemical quantities. Other initial conditions could have equally been perturbed, such as salinity, nutrients etc. Although this wasn't tested explicitly, we found with the same coupled Earth system model (GFDL ESM2M) that the choice of prognostic (interactive) versus climatological chlorophyll with identical initial conditions leads to at least qualitatively similar divergence of the model over only a few model time steps. We found that weather patterns are distinct after less than one month under such perturbations. From this we infer that any field that interacts with the dynamical state of the model can serve as an analogous perturbation to those considered here, but this is left as a topic for future testing and quantification. We also note that we used six different starting points from the preindustrial control simulation to start the 40-member ensemble simulation. Therefore the six 40-member ensemble simulation have different physical and biogeochemical states.

And of course there are many other factors limiting predictability, such as the model parameterisation. This isolation of one source of limits to prediction is appropriate but should be made clearer in the introduction, and then discussed, in the light of the results, more thoroughly in the Discussion. For example, changing BGC models, or even the remineralisation rate of organic matter, would change the time scale of AOU.
The question of how specific parameterizations within a given model impact predictability is clearly an important question, and to our knowledge this remains unexplored for marine biogeochemistry and ecosystem drivers. This has in fact been explored for the case of weather prediction (Palmer and Williams, 2008), where the inclusion of stochastic parameterizations has been argued to increase predictability. Such parameterizations are not available in the model used here, and from our understanding existing work on this topic has not considered predictions on the timescales considered here. Returning more directly to the topics raised by the reviewer, testing of the sensitivity of the biogeochemistry model will be facilitated in the future for the case of the MOM6 (isopycnal model) that will be used by both GFDL and CESM, so that one can "switch" between GFDL's newer COBALT model and CESM's Marbl/BEC model. Likewise one could explore the sensitivity to the mixing scheme (GFDL's ePBL versus CESM's KPP scheme) through its effect on the dynamical behavior of the model modulating biogeochemistry. So we anticipate that there should be a valuable opportunity to explore these questions moving forward. We have modified the caveat section in the discussion section to address this comment. See also our reply to comment 6 below.

3. [Most important point that needs addressing]. The terms "lead time" and "predictability time horizon" are used interchangeably in the last paragraph of p8, which demonstrates an inconsistency. The term lead time makes sense to me in Fig 2, 4 and 5. It is the time axis, starting at the perturbed time, along which the variability of the ensemble and controls are measured. But figures 3, 6, 7 etc. the surface plotted is labelled "lead time" when it should be "predictability time horizon". Predictability time horizon is loosely describes as PPP < 0.183 (also, is prognostic potential predictability the same thing as predictability time horizon)? But this is problematic since PPP varies with time. Should it be min t for which PPP(t) < 0.183? Check every use of lead time, PPP and predictability time horizon and make sure it is consistent in the manuscript. Predictability is also loosely defined, and would in many cases be best replaced with "predictability time horizon".
We agree with the reviewer that the terms 'lead time', 'predictability time horizon' and 'PPP' are sometimes used interchangeably in the manuscript. We have now carefully streamlined the usage of these terms throughout the manuscript. As a result, we often replaced 'predictability' with 'predictability time horizon'. In addition, we changed 'lead time' to 'predictability time horizon' in Figure 3 and Figures 6 to 13.

As the reviewer noted, the prognostic potential predictability is not equal the predictability time horizon. The prognostic potential predictability metric (i.e. PPP) varies with lead time. However, as defined in section 2.3.1, the predictability time horizon is defined as the lead time at which PPP falls below 0.183.

4. Fig. 1b. This figure would be more instructive if it was used as an example of the calculation of the PPP. If the control variance could be plotted (maybe a grey between +/- sigma) and then PPP, and the point at which PPP drops below 0.183. This would set up the rest of the manuscript better.

Many thanks for this suggestion. We have added the standard deviation (not the variance as it becomes too small to be shown) of the control simulation as horizontal dashed lines and indicate with a vertical line the predictability time horizon (i.e. when PPP falls below the predictability threshold).

[Figure]

Figure 1: Illustration of the model setup and the calculation of the predictability time horizon. (a) Simulated global mean SST of the 300-yr reference control simulation (black line) and of the six 10-yr long 40 ensemble simulations (red lines). (b) Global mean SST anomaly (i.e., deviation from the control simulation) for the ensemble simulation starting in year 170. Thick red line indicates the period over which SST is predictable (i.e. PPP ≥ 0.183), and thin red lines indicate period over which SST is unpredictable (i.e. PPP < 0.183). The dashed horizontal lines indicate one standard deviation of the control simulation and the vertical line indicates the predictability time horizon.

Also, in the caption, why do you call it the "first ensemble simulation".
This was the first set of 40 ensemble members we have performed with ESM2M, but this is not important for the reader of this manuscript. We therefore deleted the word 'first.

5. Paragraph 396 – 405 needs re-writing for clarity. I think it is trying to say that you can have different time scales for predictability for perturbations in forcing (such as anthropogenic CO2) to perturbations in initial conditions (as studied here).
We compare in this paragraph the Time of Emergence (ToE) of the marine ecosystem drivers with the predictability of these drivers. Earlier papers have found that the ToE strongly differs among the drivers. This is in stark contrast to the predictability horizon discussed in this paper, which is almost identical for all four ecosystem drivers at the global scale. We hope that the broader suite of revisions we have provided to the manuscript will clarify this message and the context in which it occurs.

6. The discussion has too much focus on obvious limitations (such as ensemble size, years started etc.) and less on more subtle limitations like time-scale of coefficients in the BGC model. The second are particularly worth of discussion here because the effort at deconvolution of the processes allows for an insightful discussion of these.
TOPAZv2 represents a hypothetically "optimal" phytoplankton physiology, i.e. it assumes that the fastest growing phytoplankton group always wins in all environments via the upper limit in growth rates as per Bissinger et al., (2008) and Eppley (1972). Similarly, because TOPAZv2 represents a steady-state ecosystem, there are no time lags between primary production and the grazing response. In the subsurface, the remineralization of particles is set to reproduce the vertical scale of the nutricline on the timescale of sinking particles, and the sinking particle velocity is fast, i.e. 100 m/day. All three factors would tend to decrease the memory associated with the real world surface ecosystem and minimize predictability. We expect coefficients relating to the longer time scales to include the dissolved organic matter remineralization and the depth scale of sinking particle remineralization, both are fairly well constrained from observations at the global scale, but we expect considerable regional and temporal variability in these. As with our comments above, quantifying these effects is outside of the scope of this study, as testing this would require an independent and more expansive site of simulations that are beyond the physical resources available for this study. We have modified the caveat section in the discussion section to address this comment.

Minor comments.
1. Whenever referring to time values, try to keep the adjectives to ones with a sense of time such as low -> short (L254), elevated -> lengthened (L195), high - > long (L391). This aids readability. There are many examples of this.
We thank the reviewer for this and changed the text accordingly.

2. The sentence L91 starting "The six "should come before the "Note" for better readability.
Changed and acknowledged.

3. L92 replace "are" with "were"?
Done.

4. Line 100. For those interpreting the equation, maybe a sentence after it "Thus the range of perturbations is evenly spread from -0.002 to 0.002 C with the control in the centre."
Many thanks for this suggestion, which we included.

5. L110 replace "underrepresented" with "underestimated"
Done.

6. Description of Eq.2 (L119-120) doesn't mention the six ensembles.
The equation is further explained on lines 125-131, where it is stated that N is the total number of different ensemble simulations (N=6) and M the number of ensemble members (M=40)

I didn't fully understand the rationale for 6 ensembles of size 40. Why not 240 members starting at all different times?
It is a common procedure for perfect modelling frameworks that different multi-member ensemble simulations started at different points in time of the control simulations are used to assess predictability. Using different points in the control simulation randomizes the initial conditions (e.g. yielding different ENSO phase states), with this intended to average across biases that may result from predictability being different across different phase of climate modes. We clarified this in section 2.2.

Also is sigma of the control the same for all ensembles?
Yes, the sigma of the control is the same for all ensembles.

Just a little bit more help here to those unfamiliar with the approach.
We hope that our answers above helped to clarify things.

7. L150 pH is approximately –log10([H+]). I know you didn't mean to define it here, but the use of (or X) sort of implies it.
We deleted the bracket to avoid confusion.

8. L184 meaning of "PPP with lead time" not clear.
We simplified it to "PPP over time".

9. L206 replace "across" with "for each of the"
We changed the sentence to ".. differences between each of the four .."

10. L234. How can a coupling enhance predictability? Sentence needs to be more carefully constructed.
We changed to sentence to: "However, biogeochemical processes lead to enhanced predictability below 500 m for $O_2$ and pH."

11. L387 "predictability of each variable".
Changed and acknowledged.

[revised manuscript text omitted]

90  biogeochemical processes relevant to assess marine ecosystem drivers at adequate temporal and spatial resolution), near perfect initial conditions and that we exclude a role for external forcing in determining or limiting predictability. Specifically, we first performed a 300-yr preindustrial control simulation (black line in Figure 1), which is branched off a pre-existing quasi-steady-state 1000-yr preindustrial control simulation. Using this 300-yr preindustrial control simulation to provide initial conditions, six 40-member ensemble simulations of 10-yr duration each are performed. Each ensemble simulation starts at different times

in the control simulation: January 1st in years 22, 64, 106, 170, 232 and 295, respectively. The six distinct initialization dates for the individual large ensemble simulations were randomly selected from the 300-yr preindustrial control simulation. This was intended to average across biases that may result from predictability being different across different phase of climate

100   modes (e.g. different El Niño Southern Oscillation phase states) within the preindustrial simulation. Note that the last ensemble exceeds the control simulation by 5 years. Each of the six ensembles consists of 40 ensemble members with micro-perturbations to oceanic initial states but with the same atmospheric, land, ocean biogeochemical, sea ice, and iceberg initial conditions. Specifically, for each ensemble member, $i$ = 1, 2, …, 40, an infinitesimal temperature perturbation $\delta$ is added to a single grid cell in the Weddell Sea at 5-m depth, similar to the approach described in Wittenberg et al. (2014a) and Palter et

105   al. (2018):

$$\delta_i = 0.0001°C \times \begin{cases} \frac{i+1}{2}: & for\ odd\ i \\ -\frac{i}{2}: & for\ even\ i \end{cases}. \tag{1}$$

Thus, the range of perturbations is evenly spread from -0.002°C to 0.002°C with the unperturbed control case in the center

[revised manuscript text omitted]

al., 2014; McGregor et al., 2014). Different physical and biogeochemical parameterizations within a given model may change
the length of the predictability time horizon. For example, TOPAZv2 represents a hypothetically optimal phytoplankton
physiology, namely the model assumes that the fastest growing phytoplankton group always wins in all environments via the
upper limit in growth rates. In addition, TOPAZv2 represents a steady-state ecosystem, such that there are no time lags between
545   primary production and the grazing response. In the subsurface, the remineralization of particles is set to reproduce the vertical
scale of the nutricline on the timescale of sinking particles, and the sinking particle velocity is fast. All three factors may tend
to decrease the memory associated with the real-world surface ecosystem and minimize predictability. For the case of weather
prediction, it has been argued that the inclusion of stochastic parametrizations increases potential predictability (Palmer and
Williams, 2008). To our knowledge, this remains unexplored for marine biogeochemistry and ecosystem drivers. In any case,
550   it would be necessary to repeat our predictability experiments with a set of different Earth system models including different
parameterizations of biogeochemical and/or physical ocean processes to investigate the dependence of our result on the model
representation (Séférian et al., 2018), in parallel with broader efforts to further evaluate noise characteristics of these models.
Additionally, the ocean model resolution of GFDL ESM2M is rather coarse and cannot represent the critical scales of small-
scale structures of circulation. Predictability studies using high resolution ocean models with improved process representations
555   are therefore needed to explore potential predictability, especially at the local scale. However, it is currently impossible in
many cases to constrain the simulated variability in biogeochemical drivers, especially for the ocean subsurface, with
observations due to limited data availability (Frölicher et al., 2016; Laufkötter et al., 2015).

Currently, no global coupled physical-biogeochemical seasonal-to-decadal forecast system is yet operational (Tommasi et al., 2017). However, our study suggests great promise that physical-biogeochemical forecast systems may have the potential to provide useful information to a wide group of stakeholders, such as, for example, for the management of fisheries (Dunn et al., 2016; Park et al., 2019). Our study therefore underscores the need to further develop integrated physical-biogeochemical forecast systems. Especially in regions with long predictability time horizons, such as the North Atlantic (for temperature, $O_2$, pH), the Southern Ocean (for temperature and $O_2$), and mid-latitudes (for NPP), installing and maintaining a spatially and temporally dense physical and biogeochemical ocean observing system would have the potential to significantly improve the effective predictability of marine ecosystem drivers.

**Appendix A**

The NPP in TOPAZ2, defined as the phytoplankton nitrogen production, is individually described for all phytoplankton groups $i$ by the product of a phytoplankton growth rate $\mu_i$ and the amount of nitrogen in the plankton group $[N]_i$:

$$\text{NPP}_i = \mu_i \cdot [N]_i. \tag{A1}$$

The growth rate of the small phytoplankton group is given by a maximum growth rate times the limiting factors of nutrients $N_{lim}$, light $L_{lim}$, and temperature $T_f$:

$$\mu_{Sm} = \frac{\mu_{max'}}{1+\zeta} \cdot N_{lim} \cdot L_{lim} \cdot T_f. \tag{A2}$$

The temperature limitation factor is:

$$T_f = \exp(k_{epp} \cdot T). \tag{A3}$$

The nutrient limitation factor is:

$$N_{lim} = \min(N_{Fe}, N_{PO_4}, N_{NO_3} + N_{NH_4}), \tag{A4}$$

with iron limitation:

$$N_{Fe} = \frac{Q_{Fe:N}^2}{Q_{Fe:N}^2 + K_{Fe:N}^2}, \quad \text{with} \quad Q_{Fe:N} = \min\left(Q_{Fe:N_{max'}} \frac{[Fe]_{Sm}}{[
[revised manuscript text omitted]

820

**Figures**

[Figure]

[Figure]

[Figure]

**Figure 1: Illustration of the model setup and the calculation of the predictability time horizon.** (a) Simulated global mean SST of the

825   300-yr reference control simulation (black line) and of the six 10-yr long 40 ensemble simulations (red lines). (b) Global mean SST anomaly

(i.e., deviation from the control simulation) for the ensemble simulation starting in year 170. Thick red line indicates the period over which

SST is predictable (i.e. PPP ≥ 0.183), and thin red lines indicate period over which SST is unpredictable (i.e. PPP < 0.183). The dashed

horizontal lines indicate one standard deviation of the control simulation and the vertical line indicates the predictability time horizon.

[revised manuscript text omitted]

**Figure C1:** (a) Simulated annual mean AMOC maximum of the 300-yr long preindustrial control simulation. The blue line indicates the 10-yr running mean. (b) Monthly mean (thin line) and annual mean (thick line) prognostic potential predictability for the AMOC maximum. The horizontal black dashed line represents the predictability threshold.

 **Table A1: TOPAZ2 parameters for small phytoplankton**

| Parameter | Value | Units | Description |
|---|---|---|---|
| $\zeta$ | 0.1 | | Photorespiration loss |
| $k_{epp}$ | 0.063 | $°C^{-1}$ | Temperature coefficient for growth |
| $\alpha$ | 2.4e-5 · 2.77e18/6.022e17 | g C (g Chl)$^{-1}$ m$^2$ W$^{-1}$ s$^{-1}$ | Light harvest coefficient |
| $\mu_{max}{'}$ | 1.5e-5 | s$^{-1}$ | Maximum growth rate at 0°C |
| $\theta_{min}^{nolim}$ | 0.01 | g Chl (g C)$^{-1}$ | Minimum Chl:C without nutrient limitation |
| $\theta_{min}^{lim}$ | 0.001 | g Chl (g C)$^{-1}$ | Minimum Chl:C with complete nutrient limitation |
| $\theta_{max}$ | 0.04 | g Chl (g C)$^{-1}$ | Maximum Chl:C |
| $K_{NO_3}$ | 2e-6 | mol N kg$^{-1}$ | NO$_3$ half-saturation coefficient |
| $K_{NH_4}$ | 2e-7 | mol N kg$^{-1}$ | NH$_4$ half-saturation coefficient |
| $K_{Fe:N}$ | 12e-6 · 106/16 | mol Fe (mol N)$^{-1}$ | Half-saturation coefficient of iron deficiency |
| $Q_{Fe:N\ max}$ | 46e-6 · 106/16 | mol Fe (mol N)$^{-1}$ | Maximum Fe:N limit |
| $Q_{P:N\ max}$ | 0.1458 | mol P (mol N)$^{-1}$ | Maximum P:N limit |

**Seite 38: [1] Gelöscht**                    **Thomas Frölicher**                    **09.03.20 10:34:00**

**Seite 38: [1] Gelöscht**                    **Thomas Frölicher**                    **09.03.20 10:34:00**

**Seite 38: [1] Gelöscht**                    **Thomas Frölicher**                    **09.03.20 10:34:00**

**Seite 38: [1] Gelöscht**                    **Thomas Frölicher**                    **09.03.20 10:34:00**